Evidence that ebolaviruses and cuevaviruses have been diverging from marburgviruses since the Miocene

Taylor Derek J. djtaylor@buffalo.edu
Ballinger Matthew J.
Zhan Jack J.
Hanzly Laura E.
Bruenn Jeremy A.
Department of Biological Sciences, The State University of New York at Buffalo , Buffalo, NY , USA
Wilke Claus
Electronic publication date: 2014 Sep 2
Publication date: 2014
Volume: 2
Electronic Location ID: e556
Received 2014 Jul 16; Accepted 2014 Aug 12
Copyright: © 2014 Taylor et al.
Copyright year: 2014
Copyright holder: Taylor et al.
License: This is an open access article distributed under the terms of the Creative Commons Attribution License, which permits unrestricted use, distribution, reproduction and adaptation in any medium and for any purpose provided that it is properly attributed. For attribution, the original author(s), title, publication source (PeerJ) and either DOI or URL of the article must be cited.
License URL: https://creativecommons.org/licenses/by/4.0/

Keywords: Ebolavirus, Marburgvirus, Paleovirus, Cricetidae, VP35, NP, Filoviruses, Divergence estimation

Funding: The author declares there was no funding for this work.

==============================
An understanding of the timescale of evolution is critical for comparative virology but remains elusive for many RNA viruses. Age estimates based on mutation rates can severely underestimate divergences for ancient viral genes that are evolving under strong purifying selection. Paleoviral dating, however, can provide minimum age estimates for ancient divergence, but few orthologous paleoviruses are known within clades of extant viruses. For example, ebolaviruses and marburgviruses are well-studied mammalian pathogens, but their comparative biology is difficult to interpret because the existing estimates of divergence are controversial. Here we provide evidence that paleoviral elements of two genes (ebolavirus-like VP35 and NP) in cricetid rodent genomes originated after the divergence of ebolaviruses and cuevaviruses from marburgviruses. We provide evidence of orthology by identifying common paleoviral insertion sites among the rodent genomes. Our findings indicate that ebolaviruses and cuevaviruses have been diverging from marburgviruses since the early Miocene.

Introduction

Knowledge of the timescale of evolution is a critical part of understanding host-virus interactions. Studies of viral systems that have evolved for tens of millions of years would perhaps be complicated by host shifts, broad geographic distributions, and functional divergences. Knowledge of divergence times might also affect design of vaccines and programs that identify emerging pathogens. However, the timescale of viral evolution has remained controversial (Gilbert & Feschotte, 2010; Holmes, 2003; Patel, Emerman & Malik, 2011; Sharp & Simmonds, 2011; Wertheim & Kosakovsky Pond, 2011). Fossil and geographic calibrations are normally absent and evolutionary rates based on isolation dates of historical strains often grossly underestimate long-term divergences. Part of the underestimation is due to the failure of commonly used models to accommodate the strong purifying selection of viral proteins (Duchêne, Holmes & Ho, 2014; Patel, Emerman & Malik, 2011; Wertheim & Kosakovsky Pond, 2011). But other aspects of viruses such as variation in replication rate also affect clock-based estimates (Hicks & Duffy, 2014; Holmes, 2003). Even models that accommodate purifying selection will eventually encounter a mutational saturation problem (Wertheim & Kosakovsky Pond, 2011). Age estimation using co-phylogeny with the host seems more promising, but detailed co-phylogenies are still uncommon and can be complicated by host jumping (Holmes, 2003).

Another potentially reliable source of minimum divergence times is endogenous paleoviral elements (Katzourakis et al., 2007). Over the last decade, evidence of the most unexpected class of paleoviral elements, Non-retroviral Endogenous RNA Viral Elements (NERVEs), has been provided for each major eukaryotic group by sequencing across the integration boundaries of putative viral elements and host genomes (Crochu et al., 2004; Horie et al., 2010; Liu et al., 2010; Tanne & Sela, 2005; Taylor & Bruenn, 2009; Taylor, Leach & Bruenn, 2010). BLAST searches of animal genome databases alone suggest that representatives of all known viral genome architectures are involved in the formation of paleoviral elements (Belyi, Levine & Skalka, 2010; Katzourakis & Gifford, 2010). Agreement of the NERVE phylogeny with the host phylogeny is evidence of insertion in the genome of a common ancestor. This pattern can be complicated by the formation of non-orthologous copies from independent insertions, duplications and horizontal transfers (Taylor & Bruenn, 2009). But, these complications become less important in phylogenies with greater taxonomic representation. Even stronger support for orthology is provided by evidence of common integration sites for NERVEs (Katzourakis & Gifford, 2010; Taylor, Leach & Bruenn, 2010). If the host genomic flanking sequences show significant similarity (microsynteny), then it is unlikely that NERVE insertions are independent (given the large number of possible insertions sites in eukaryotic genomes). Ballinger et al. (2014), for example, were able to identify microsyntenous NERVEs from a novel bunyavirid for the genus Drosophila and estimate a minimum date of 42 MY. As with the hosts, the paleoviral sequences are “sister” phylogenetic groups supporting a single origin. So, the strongest paleoviral calibrations satisfy two conditions: evidence of a common integration site in the host genomes and of a similar phylogeny of host and paleovirus.

Age estimation by synteny has limitations. As with real fossils, the age estimates based on paleoviruses are minimum dates based on the material currently available. The synteny of the oldest copies may be difficult to establish because of chromosomal evolution that occurred post-integration. An additional source of uncertainty arises from the dating of actual host fossils and of host divergences. It also may be imagined that NERVE-virus phylogenetic comparisons suffer because the mutation rate of RNA viruses is orders of magnitude greater than the mutation rate of the hosts. However, this reasoning ignores the growing evidence of strong purifying selection in viruses—a high mutation rate is not necessarily reflected in the amino acid substitution rate. Indeed, ancient NERVEs would simply be undetectable for viruses that diverged rapidly at the amino acid level.

Still, very few paleoviral calibrations are available for internal nodes of phylogenies of extant RNA viruses. Taylor et al. (2011) reported that the family Filoviridae must be at least 13 MY old because fossil copies of the NP and VP35-like genes have been integrated at common sites shared among the mouse (Mus musculus) and the rat (Rattus norvegicus) NCBI reference genomes. However, paleoviral calibrations that would permit estimation of a minimum divergence date for extant ebolaviruses, cuevaviruses and marburgviruses are unknown. Molecular estimates of the age of the common ancestor of extant known filovirids fall into two time ranges. One range is coincident with the rise of agriculture in humans from 7,100 to 10,400 years ago (Carroll et al., 2013; Suzuki & Gojobori, 1997). The other range is from the Middle Pleistocene at 155,000 years ago (Negredo et al., 2011). Still others have stated that the oldest extant filovirids (and other RNA viruses) are in a divergence zone that is simply recalcitrant to molecular clock dating (Wertheim & Kosakovsky Pond, 2011). In such a zone, even models that have been corrected for purifying selection will fail to a temporal signal that has been destroyed by synonymous substitutions.

Here we show that the limitations for using the molecular clock to date RNA viruses can be mitigated by the discovery and dating of orthologous paleoviral elements within clades of extant RNA viruses. From congruent evidence of two genes we report that the divergence of the known extant filovirids (marburgviruses, ebolaviruses, and cuevaviruses) is likely older than the Miocene ancestor of the hamsters and voles—a separation that is orders of magnitude greater than previous Holocene and Middle Pleistocene estimates of divergence.

Materials & Methods

PCR and DNA sequencing

We obtained a tail clipping of a dead specimen of a meadow vole from western New York state. We extracted DNA using the Epicentre Quickextract kit. PCR reactions with primers based on the genome assembly of Microtus ochrogaster (Wagner, 1842) were designed to amplify from the exon across the VP35-like gene insert boundary and from the intergenic region across the NP-like boundary of putative orthologs of hamster loci. Primers based on the assembly of the genome of the prairie vole (Microtus ochrogaster) for the VP-35-like region were: GAGCAGGCTTTTGCTTTGATTCCAG (forward), CTGATCTCAGCTATCTCACCTGCTAAGA (reverse). For the NP-like region primers were: TGCATTGCTTGGCCGTTCTGTATGC (forward) and ATAAGACATGCTCCTTGTCTTGAAG (reverse). The 5′ end of the mitochondrial COI gene region was also PCR amplified using custom primers based on published sequences of Microtus: TTACAGTCTAATGCTTTACTCAGCC (forward), ACTTCTGGGTGTCCGAAGAATCAG (reverse). PCR products were purified and submitted for Sanger sequencing at the Roswell Park Cancer Institute’s Biopolymer facility. Chromatograms were assembled and trimmed using Geneious 7.0 (Biomatters).

Bioinformatics

Genomic sequences from the NCBI WGS and reference genome databases were obtained by using protein sequences of NP and VP35 of Ebola virus as queries. We used the tBLASTn algorithm with mammals as a taxonomic delimiter. Resulting contigs with an expect value <10−5 were retained and exported. The NP and VP35 protein sequences from Ebola virus were then used to search for significant matches of the contigs using the FASTA program tfasty (Pearson, 2004). Translated sequences were then prepared as a FASTA format alignment by changing the NCBI header to a user-friendly name with a Python script. Sequences were submitted to the E-INS-I algorithm of MAFFT for multiple sequence alignment (Katoh & Standley, 2014). The resulting alignment file was then submitted to the transitive consistency score (TCS) algorithm of T-Coffee to assess alignment reliability (Chang, Di Tommaso & Notredame, 2014). Unfiltered and filtered alignment files in FASTA format for each filovirid-like gene are provided in the Supplemental Information. The lowest scoring categories of columns were successively filtered from the alignments using a Python script to assess the effect of rapidly evolving or differently evolving sites on branch support. To assess possible effects of increased rate evolution at the tips of the tree, the sequences of ancestral nodes of endogenous viral clades were estimated using the three ancestral reconstruction methods (Delport et al., 2010) in HyPhy (joint maximum likelihood, marginal maximum likelihood, and mode of the posterior distribution of characters) with a JTT + gamma substitution model. Midpoint and outgroup rooting was carried out in Figtree 1.4, with the outgroup being clades of mammalian filovirid-like NERVEs outside of the clade of extant known filovirids. Seaview 4.5.2 (Gouy, Guindon & Gascuel, 2010) was further used to explore rooting the tree “at the point in the tree that minimizes the variance of root-to-tip distances”. Protein models were fit to the resulting alignments using Partitionfinder protein (Lanfear et al., 2012). Phylograms were estimated using Bayesian MCMC using MrBayes 3.2.2 (Ronquist et al., 2012) as implemented at the CIPRES science gateway (Miller, Pfeiffer & Schwartz, 2010). Priors for MrBayes included the amino acid model fixed as JTT (Jones). The sampling frequency was every 1000 generations with the MCMC analysis continuing until the average standard deviation of split frequencies was less than 0.01. We used a burnin fraction of 0.25 and a random starting tree. Branch reliability was assessed with Bayesian posterior probability values and by approximate likelihood ratio tests (aLRT). Maximum likelihood was carried out in PhyML 3.1 (Guindon et al., 2009) as implemented in Seaview 4.5.2 with the subtree pruning and regrafting search algorithm (SPR). Phylograms were visualized in Figtree 4.1 (Rambaut, 2012) and Adobe Illustrator.

Microsynteny of NERVE insertion sites was assessed by carrying out a BLAST search with NERVEs as the queries and the annotated reference genomes of rodents as the databases. NERVE-containing segments were compared among rodents after using the progressive alignment algorithm in the Mauve (Darling, Mau & Perna, 2010) plugin of Geneious 7 (Biomatters). Patristic genetic distances (measured from branches on a gene tree) based on nucleotide alignments of filovirid-like regions in rodents and their extracted intronic or intergenic backgrounds were estimated in Seaview 4.5.2 (Gouy, Guindon & Gascuel, 2010) using the HKY distance (Hasegawa, Kishino & Yano, 1985) with site rate variation being optimized in four categories.

Results and Discussion

Significant Blast expect values were found for 50 NP-like sequences from mammalian genomes and 11 VP35-like sequences. Only one assembly per species was retained for the analysis. We detected several previously unknown filovirid-like NERVEs from rodent genomes. These included NERVEs from the Upper Galilee Mountains blind mole rat (Spalax galili), the golden hamster (Mesocricetus auratus), the prairie vole (Microtus ochrogaster), and the North American Deermouse (Peromyscus maniculatus bairdii). An additional NERVE sequence was amplified by PCR from the meadow vole (Microtus pennsylvanicus). Our cytochrome c oxidase subunit 1 mitochondrial sequence is consistent with the taxonomic identification as it yielded a 99% identity score with sequences from meadow voles (e.g., JQ350481.1). Additional known filovirid-like sequences from mouse and rat genomes and those present in EST libraries or isolated from mammalian genomes by PCR (Taylor, Leach & Bruenn, 2010; Taylor et al., 2011) were excluded because we focused on the relationships within the clade of known extant filovirids.

Both the NP-like (Fig. 1) and the VP35-like (Fig. 2) sequence phylogenies revealed a clade of cricetid rodent sequences within the clade of extant filovirids. Indeed, both genes had cricetid clades paired with ebolaviruses and cuevaviruses to the exclusion of marburgviruses. Taylor et al. (2011) had previously identified this position for the genomes of a single rodent, the striped dwarf hamster (Cricetulus barabensis griseus), but here we have found support for sequences of other cricetid rodents forming a monophyletic clade. This is the most closely related clade of endogenous mammalian genes known for filovirids. The phylogenetic positions are strongly supported by posterior probabilities and aLRTs.

Figure 1 Phylogenetic relationships of filovirid NP-like paleoviruses in mammalian genomes and amino acid sequences from extant filovirids.

Bayesian posterior probabilities for the extant filovirus clade greater than 0.95 are shown as black circles. The phylogeny is based on an alignment with transitive consistency scores <3 filtered. The Blue colors represent branches leading to rodent sequences. Red colors represent branches leading to extant viral sequences. Black bars represent branches leading to non-rodent mammalian sequences. Taxonomic labels indicate phylogenetic placement of sequences from specimens assigned to the given taxon.

Figure 2 Phylogenetic relationships of filovirid VP35-like paleoviruses in mammalian genomes and amino acid sequences from extant filovirids.

Bayesian posterior probabilities for the extant filovirus clade greater than 0.95 are shown as black circles. The phylogeny is based on an alignment with transitive consistency scores <3 filtered. Blue colors represent branches leading to rodent sequences. Red colors represent branches leading to extant viral sequences. Black bars represent branches leading to non-rodent mammalian sequences. Taxonomic labels indicate phylogenetic placement of sequences from specimens assigned to the given taxon.

The occurrences are unlikely to be assembly artifacts because the genomes in question are NCBI reference genomes with strong sequence coverage. The striped dwarf hamster (C. griseus) has independent genome assemblies that agree on the insert locations. Also, the only mammalian species in this filovirid clade are cricetid rodents, some of which have identical insertion locations in their genomes. The pattern of shared insertion among monophyletic taxa is a prediction of common ancestry rather than of assembly artifacts. Finally, we carried out PCR in the meadow vole using primers designed to flank the VP35-like region of the prairie vole (which has a genome project). The PCR reaction was positive and the sequence had strong identity to the microtine sequence from the genome assembly (Fig. 3). Excluding indels the sequence across this putative insert region showed 94% (583 nt) identity between the genome assembly of the pairie vole (M. ochrogaster) and the PCR product of the meadow vole (M. pennsylvanicus). Our partial sequence is consistent with an orthologous insert of a VP35-like sequence in the genome of the meadow vole (M. pennsylvanicus) and confirms the assembled location of this region in rodents in the 3′ intron of the Tax1-binding protein 1 (TAX1BP1) gene locus.

Figure 3 DNA sequence validation of integration for the filoviral VP35-like sequence in voles of the genus Microtus.

The section of the intron common to rodent introns is highlighted in gray and the proposed filovirid-like insert is highlighted in red. Sequence comparisons (colored blocks are differences) between the PCR product (black bar) for the meadow vole (M. pennsylvanicus) and the genome assembly for the prairie vole (M. ochrogaster) are shown for (A) the shared intron of rodent genomes and (B) the proposed insert containing a filovirid VP35-like sequence in genomes of cricetid rodents.

We explored the possibility of systematic error contributing to the pairing of cricetid sequences with ebolaviruses. Long branch attraction (LBA) can occur with real data even under model-based approaches that account for among-site rate variation (Anderson & Swofford, 2004; Omilian & Taylor, 2001; Taylor & Piel, 2004). In some cases, distant outgroups can play a role in LBA (Sanderson et al., 2000). It is expected that support for LBA groupings will be reduced if sites that are rapidly evolving or that lack agreement among pairwise alignments are reduced in the data. However, with the VP35-like and NP-like genes, successively filtering such sites according to the transitive consistency score either increased support for the observed cricetid/ebolavirus pairing or had no effect on support (Fig. 4). Support for the internal position eroded only when the number of sites had been reduced to less than 15% of the data for the NP gene and 34% for the VP35 gene. We also note that similar LBA conditions between the genes are lacking because the branch length patterns are reversed for the two genes. For the VP35 phylogeny, marburgviruses have the longest distance to the root among extant viruses, but for the NP gene, ebolaviruses have the longest distance to the root. Yet, in each case the cricetid sequences group with ebolaviruses and cuevaviruses. The increase in support for this clade with filtering, then, is most likely a result of culling evolutionary noise from the mammalian genes. Most of the NERVEs are pseudogenes that accumulate evolutionary noise in the form of indels and reading frame disruptions.

Figure 4 Graphs of phylogenetic support values for the branch that groups rodent sequences with ebolaviruses and cuevaviruses to the exclusion of marburgviruses.

(A) the NP-like region and (B) the VP35-like region. The x-axis represents the size of the alignment after culling sites according to their transitive consistency scores (TCS). Note that successive removal of the sites that most disagree among pairwise alignments fails to erode support for the branch in question until the alignment size is small. aLRTs are approximate likelihood ratio tests.

To further explore a role for outgroups in affecting the relationships of the ingroup we carried out several analyses with new alignments that omitted outgroup taxa. Every analysis for the VP35-like gene alignment grouped the cricetid sequences inside the clade of extant filovirids with strong support (Table 1). The ingroup analysis using the complete NP-like alignment indicated that cricetid sequences grouped outside of the extant filovirids. However, replacing the cricetid clade with ancestral reconstructions of this clade of pseudogenes moved the cricetid NP-like sequences internal to the extant filovirids. Ancestral reconstruction of the NERVE clade reduced gaps that may have contributed to a biased attraction of extant viral genera with fewer indels. In support of this notion, removing the most rapidly evolving sites and using ancestral reconstructions of the cricetid copies gave the same results as the outgroup-rooted sequences for both genes. So, phylogenetic analysis of ingroup sequences alone after reducing the most rapidly evolving and rapidly eroding sites (including indels) indicates that the position of the cricetid sequences within extant filovirids is unlikely to be the result of an outgroup bias.

Table 1 Exploration of rooting, taxon set, and filtering of rapidly evolving sites on the phylogenetic position of filovirid-like endogenous genes in cricetid rodents.

Likelihood scores and aLRT branch support values are given for the best topologies found. Observed ingroup topologies are shown in Newick format (brackets and commas) where E stands for ebolaviruses, L for Lloviu virus, C for cricetid filovirid-like and M for marburgvirus sequences.

Alignment	Sites
remaining	Rooting	((E, L, C), M)	((E, L), (C, M))	((E, L, M), C)	
VP35	390	Outgroup, midpoint, and minimum variance	−8451.3 (0.60)			
NP	991	Outgroup, midpoint, and minimum variance	−40584.3 (0.96)			
VP35 (TCS filtered)	237	Outgroup, midpoint, and minimum variance	−5360.3 (0.78)			
NP (TCS filtered)	391	Outgroup, midpoint, and minimum variance	−22024.7 (0.98)			
VP35 ingroup	381	Minimum variance	−6468.8 (1.0)			
VP35 ingroup (TCS filtered)	280	Minimum variance	−5061.6 (0.96)			
NP ingroup	858	Minimum variance			−19899.3 (1.0)	
NP ingroup (TCS filtered)	487	Minimum variance			−10813.9 (1.0)	
VP35 ingroup ancestral NIRV	381	Minimum variance		−10813.9 48 (0.99)		
NP ingroup ancestral NIRV	858	Minimum variance		−10834.9 (1.0)		
VP35 ingroup ancestral NIRV (rapid sites filtered)	295	Minimum variance	−3648.6 (1.0)			
NP ingroup ancestral NIRV(rapid sites filtered)	534	Minimum variance	−5692.5 (1.0)			

Microsynteny was observed among a monophyletic clade of VP35-like NERVEs in cricetid rodents. The (Microtus, (Cricetulus, Mesocricetus)) grouping agrees with rodent taxonomy, with the microsynteny being apparent at the genic and the nucleotide level. Namely, these cricetid rodents share a VP35-like insert in the 3′-most intron of the Tax1-binding protein 1 (TAX1BP1) gene locus (Fig. 5). This insertion site is also identical to that of the partial VP35-like gene from the meadow vole (M. pennsylvanicus) that we amplified by PCR. TAX1BP1 is involved in the down regulation of inflammation genes (Verstrepen et al., 2011). Interestingly, both the TAX1BP1 of mammals (Parvatiyar, Barber & Harhaj, 2010) and VP35 of ebolaviruses (Basler et al., 2003; Hartman, Towner & Nichol, 2004; Hartman et al., 2008) inhibit IRF (interferon regulatory factor)3, a critical transcription factor for the initiation of viral innate immunity in mammals. The host’s inhibition of the interferon pathway is necessary to prevent the deleterious effects of inflammation. A C-terminal motif (PRACQKSLR) (Hartman, Towner & Nichol, 2004; Hartman et al., 2008) of the VP35 of ebolaviruses targets the same transcription factor to inhibit mammalian innate immunity to viruses. Unlike the more divergent VP35-like NERVEs from bats (Belyi, Levine & Skalka, 2010), the cricetid VP35-like NERVEs show evidence of conservation of the three basic residues affecting the interferon response. Specifically, two of the striped dwarf hamster (C. griseus) NERVEs and one of the ancestral reconstructions have these important residues (PRPCQKSIR). The shared insertion of a viral gene that inhibits IFR3 in a mammalian gene that plays a key role in the inhibition of IFR3 immediately suggests selective maintenance of the integration. The VP35 of Ebola virus fails to inhibit the type I interferon response in hamsters (Ebihara et al., 2013). Also, Ebola virus infections in hamsters cause downregulation of proinflammatory cytokines, while still inducing a type I interferon response (Ebihara et al., 2013). Although intron sequence variation is known to be functional in mammals (Praetorius et al., 2013), we are unaware of functional studies of the VP35-interrupted TAX1BP1 of rodents. Functional studies are needed to address the possibility that co-option of a viral interferon pathway regulator (VP35) contributes to the immune response of hamsters.

Figure 5 Cartoons comparing an orthologous genomic region among rodents that (A) lack a filovirid VP35-like insert; and (B) possess an orthologous filovirid VP35-like insert.

Closeups of the upstream and downstream putative insertion boundaries are shown revealing microsynteny at the nucleotide level (the blocks of differing colors are nucleotides). A gray bar represents intronic sequence and a red bar represents the putative filovirid VP35-like insert region.

Microsynteny was also observed for the NP-like genes of cricetid rodents (Fig. 6). The oldest case of microsynteny involved inserts in the same intergenic region between the gliomedin and cytochrome P450 19A1-like loci shared by rodents of the following genera: Microtus, Mesocricetus, and Cricetulus. Hamsters of the genera Cricetulus and Mesocricetus had very high similarity of flanking sequences (Fig. 6) indicating a shared insert site. The insert in the vole genome was just under 10 kbp from the hamster insert site. Possible causes of the differing insert location could be a modest rearrangement, an assembly artifact, an independent insert in the same intergenic region, or tandem duplication followed by loss of the original insert. We sequenced across the putative insert boundary using DNA from the meadow vole (M. pennsylvanicus) as a template to assess the assembly. The assembly and location of the insert was verified by the sequence, which had at least 91% nucleotide sequence identity with the assembly of the pairie vole (M. ochrogaster). We then compared the pairwise sequence divergences of putatively orthologous NERVEs, under the expectation that orthologous NERVEs would evolve at a similar or slower rate than the background intron or intergenic region of their insertion. The NP-like insert of the vole evolved at about twice the rates of the intronic, intergenic and other NERVE comparisons, suggesting that the vole insert may not be orthologous to the hamster insert (Fig. 7). A dotplot comparison of this sequence revealed no evidence for recombination. We conclude that there is strong evidence for the orthology of NP-like inserts for hamsters of the genera Cricetulus and Mesocricetus in the intergenic region between gliomedin and cytochrome P450 19A1-like loci. However, the NP-like insert of the vole may be an independent insert or a paralog. If true, then gene order evidence alone for mammals may be too crude a measure to evaluate orthology for NERVEs.

Figure 6 Cartoon comparing the aligned genomic regions of cricetid rodents that contain a putative orthologous filovirid NP-like sequence.

For this genomic segment, the assembly from the North American deermouse (Peromyscus manipulates bairdi) lacks a detectable insert. Specimens of Microtus ochrogaster, Cricetulus griseus, and Mesocricetus auratus share an insert in the same intergenic region. However, the insert of the prairie vole (M. ochrogaster) is about 10 kbp upstream of the shared insert of members of the other species. The identity bar reveals strong sequence similarity in this intergenic region and flanking the insert site of the striped dwarf hamster (C. griseus) and the golden hamster (M. auratus). Most non-identity is due to indels in the intergenic region.

Figure 7 A graph comparing genetic distance (patristic) among putatively orthologous filovirid-like gene inserts in cricetid rodent genomes.

(A) compares pairwise distances for the VP35-like inserts with those based on the intronic background. (B) compares distances of the NP-like inserts with those of their intergenic sequence background.

The combination of monophyly, phylogenetic agreement with rodent subfamilies, topological agreement between genes, and microsynteny indicate that the cricetid rodent VP35-like insertion was present in the common ancestor of hamsters and voles and the NP-like insertion was present in the common ancestor of hamsters of the genera Cricetulus and Mesocricetus. It may be argued that integrated viruses evolve by a different mode than do extant viruses or that integrated viruses are too distant from extant viruses to be biologically relevant. However, strong purifying selection has been demonstrated to occur in both RNA viruses (Wertheim & Kosakovsky Pond, 2011) and in their integrated eukaryotic versions (Taylor et al., 2011). Some of these show both RNA expression products and purifying selection (Ballinger et al., 2014). In at least one case an integrated RNA viral gene produces a protein product (Taylor et al., 2013). Under strong purifying selection, similarity at the amino acid level can be preserved despite differences in the mutation rates for many millions of years. Our analysis of the transient consistency scores indicates that if differing modes of evolution exist, they have little effect on the major relationships of the sequences from inferred amino acids. Given our evidence for orthology of the ebolavirus-like genes, we can provide a minimum estimate of the age of the insert as the age of the common ancestor of hamsters and voles. Molecular clock estimates using fossil calibrations agree that hamsters and voles had a common ancestor in the Miocene (Abramson et al., 2009; Fabre et al., 2012; Horn et al., 2011; Jansa, Barker & Heaney, 2006; Parada et al., 2013; Steppan, Adkins & Anderson, 2004). Indeed, these studies indicate a divergence date of about 18 MY ago (error bars span much of the Miocene). The common ancestor of hamsters of the genera Cricetulus and Mesocricetus has also been estimated in the Miocene at 7–12 MY ago (Neumann et al., 2006). If the phylogenetic placement of the cricetid NERVEs within known extant filovirids is correct, then the divergence of marburgviruses from other filovirids (ebolaviruses and cuevaviruses) must also be at least as old as the Miocene. This age is orders of magnitude older than previously thought and will likely aid in understanding the comparative biology of filovirids. The differing genome architecture, transcriptional editing, and immunological reactivity of cuevavirions, ebolavirions, and marburgvirions (Kuhn et al., 2010) had a much longer time to evolve than the rise of agriculture. Our results provide strong evidence that molecular clock based estimates for extant filovirids have been severely underestimated as predicted by the saturation problem posed by Wertheim & Kosakovsky Pond (2011). However, our methods also provide a solution to this problem—dating of orthologous paleoviruses.

Small rodents appear overrepresented amongst the NERVEs of filovirids. The analysis here further bolsters this pattern with the mouse-related clade being represented in each of the three deep clades of filovirids (fossil and extant). Indeed, the genome of the North American deermouse (P. maniculatus bairdii) has NERVEs from all three major filovirid clades, suggesting multiple historical integrations of divergent filoviral lineages. It is unknown why the genomes of mouse-like rodents appear overrepresented in the list of mammals with filovirid-like NERVEs. Because NERVEs originate as rare macromutations, nearly all inserts will be lost by failing to achieve integration into germ-line cells. Those that become endogenous will most likely disappear after genetic drift or selection. Casual infection alone, then, is unlikely to result in the fixation of numerous long-lived integrations in rodents (Johnson, 2010). The evidence of repeated genomic and evolutionary interactions of filovirids (including the extant clade of filovirids) with cricetid rodents should be considered when comparing the differing immunological responses among mammals to infections with modern filovirids (Wahl-Jensen et al., 2012).

Cricetid rodents have captured orthologous NP and VP35-like gene segments from filovirids that group phylogenetically within the extant filovirids. The sharing of these genomic sections provides the first evidence that extant known filovirids have been diverging since the Miocene. The results show that fossil copies of RNA viruses can provide minimum estimates of divergence in a divergence range that is recalcitrant to present molecular clock methods with extant viruses. Our results also bolster evidence that mouse-like rodents have had repeated genomic interactions with filovirids. Our finding of a filoviral insert that interrupts an important regulator of the innate antiviral response also informs hypotheses regarding the possible biological significance to the mammalian host of such inserts.

Supplemental Information

Supplemental Information 1 Multiple sequence alignment files of filovirus-like sequences in mammals in fast format

NP-like and VP35-like alignment files from mammals and viruses. Stop codons are treated as gaps.

Click here for additional data file.

We thank Dr. Solon Morse for the tail clipping of the meadow vole.

Additional Information and Declarations

Competing Interests

Author Contributions

DNA Deposition

Jeremy Bruenn is an Academic Editor for PeerJ. The authors declare there are no competing interests.

Derek J. Taylor conceived and designed the experiments, analyzed the data, contributed reagents/materials/analysis tools, wrote the paper, prepared figures and/or tables, reviewed drafts of the paper.

Matthew J. Ballinger and Laura E. Hanzly conceived and designed the experiments, performed the experiments, analyzed the data, reviewed drafts of the paper.

Jack J. Zhan performed the experiments, analyzed the data, contributed reagents/materials/analysis tools, prepared figures and/or tables, reviewed drafts of the paper.

Jeremy A. Bruenn analyzed the data, reviewed drafts of the paper.

The following information was supplied regarding the deposition of DNA sequences:

GenBank KM189810–KM189812.

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
