# Peer review of "Evidence that ebolaviruses and cuevaviruses have been diverging from marburgviruses since the Miocene"

_PeerJ, doi:10.7717/peerj.556_

## Round 0.1 · original submission · Minor Revisions

· Academic Editor

Minor Revisions

The reviewers mostly point out minor issues of presentation that should be straightforward to address.

Reviewer 1 ·

Basic reporting

Only one complaint regarding citations.

Experimental design

Very clear exposition of the methods and assumptions in the model.

Validity of the findings

Very clear and impactful findings. Of high interest to the community, especially the observation of the rodent filoviral NIRVs.

Additional comments

I really liked this paper and enjoyed reading it immensely. For the most part, I found the logic, and the experiments really easy to follow- well reasoned. However, there is one rather jarring omission. To my mind, the problem of an apparent discrepancy between endogenous virus based estimates and current-day virus based estimates was first highlighted by a very important paper from Feschotte and Gilbert (PLOS Biology) on hepadnaviruses in birds. Indeed, this was some of the motivation for the Holmes 2003 review as well as the review by Patel et al. Curr Opin Virol (2012) which both discussed this at length. I feel that omitting citation to this paper is unacceptable (disclosure: I am not one of the authors) and indeed, citing some papers/reviews and not others is perhaps not ideal.

My second minor comment is that I think Joel would very strongly flinch at the "Wertheim zone". This term should not be used particularly because things like the "Felsenstein zone" etc have essentially so much ambiguity they lack any precision.

Reviewer 2 ·

Basic reporting

No comments/complaints. The article is written in clear and professional English, is organized logically, and properly referenced. The introduction/background is to the point, refreshingly without diverging into the usual hyperbole associated with many filovirus-related manuscripts.

Experimental design

No comments/complaints. The article is original and of importance to filovirology. The research question is clearly described and methods are well chosen and properly outlined. Reproducibility should not be a problem if the same parameters and datasets are used.

Validity of the findings

No comments/complaints. The data seem robust, well controlled, and statistically sound (although the conclusions may change drastically in the future once more and more complete mammalian genomes become available or novel filoviruses are discovered). The authors take great care not to be hyperbolic.

Additional comments

The article describes a well-controlled, carefully performed phylogenetic analysis using filovirus genes and mammalian endogenous filovirus gene-like RNA elements. Using the currently available mammalian genome datasets, the authors come to the conclusion that integration of ebolavirus-like and cuevavirus-like genes into cricetid rodent genomes occurred after the divergence of ebolavirus/cuevaviruses from marburgviruses – and thereby that filoviruses are orders of magnitudes older than previously thoughts. I am impressed by the results of the study, enjoyed reading the manuscript, and am sure it will be a useful addition to the expanding field of filovirology. I have only minor concerns that I hope the authors would be willing to address to further improve their work. These are almost exclusively stylistic in nature.

1. Title: “Evidence that ebolavirus and cuevaviruses have been…”would be more precise given the described results. I would also suggest the authors carefully screen the article and insert “/cuevaviruses” where appropriate/clarifying (including the abstract).

2. Abstract and elsewhere: “paleoviruses of two genes” is grammatically awkward. A “paleovirus” would be a virus that has once existed but does not anymore. Genes of that virus that have been integrated into a genome would be “two paleovirus genes” or “two genes of a paleovirus”. I would suggest the authors carefully screen the article for the occurrence of “paleovirus” and modify the sentences accordingly.

3. Introduction:
- Line 9: “Evolutionary timescales might also affect…” is grammatically awkward. Timescales can’t affect anything. They just are. I would suggest rephrasing.

- Line 22: “Several…(NIRV’s) have been isolated…”: is that true? Are there actual isolates, i.e. viruses growing in culture, described in all the cited papers? If not, I would suggest being more precise (“have been detected/sequenced…”). In line with the comment above I would also suggest re-evaluating the use of “NIRV”. Integrated “viruses” would mean that full or close to complete genomes of viruses can be found, rather than just parts of them. The often uses “endogenous elements” nomenclature seems to be more appropriate.

- Line 35 etc.: I commend the authors for using the taxon-specific suffix nomenclature of Vetten and Haenni (“bunyavirid” and “filovirid”). However, the authors should assure consistency and therefore screen for the occurrence of “filovirus” and replace those instances with “filovirid” (or replace all “filovirid” with “filovirus”). For instance, line 121.

- Line 51: “between” should probably be “shared among”.

- Line 51” “the mouse and the rat genomes” is very unspecific. Which mouse and which rat?

- Line 55 etc.: is “YA” a common abbreviation?

- Line 63-64: genera are capitalized and italicized; their member groups are written without italics in lower case and are by definition plural. Therefore the sentence should either be “…known extant filovirus genera (_Marburgvirus, _Ebolavirus, and _Cuevavirus)” [with “_” denoting that these words should be italicized] or “known extant filoviruses (marburgviruses, ebolaviruses, and cuevaviruses)”. Because genera are taxa, and because viral taxa are concepts and not real things, genera do not really exist. The latter version would therefore be preferable.

- Line 65 same problem: subfamilies are concepts and therefore don’t have ancestors. A better alternative would be “hamster and voles”

4. Bioinformatics
- Line 86, 88, and elsewhere: _Zaire ebolavirus is a species. Its member virus is called “Ebola virus”. Sequences are from viruses, not from species. Therefore, “Zaire ebolavirus” should be replaced with “Ebola virus” throughout the manuscript (note that Ebola virus is one of five ebolaviruses. i.e. the existence or absence of the space and presence or absence of capitalization are important for meaning).

5. Results and discussion
- I would strongly suggest the authors use a consistent vernacular mammalian nomenclature throughout the manuscript. The standard reference work for vernacular names of mammalian species members is Wilson & Reeder’s Mammals of the World (the database can be accessed at http://www.departments.bucknell.edu/biology/resources/msw3/). Accordingly, it should be “Upper Galilee Mountains blind mole rat (_Spalax _galili)” [the authors write _Nannospalax but _Spalax takes priority]; “North American deermouse” [no space in “deermouse”] etc.

- Line 133: a species does not have a sequence. Only its members do. Hence please replace _Microtus pennsylvanicus” with “meadow vole”. Likewise line 142: should be “striped dwarf hamster (_Cricetulus _barabensis _griseus)”; line 271 should be “of North American deermice (_P. _maniculatus _ bairdii)”

- Line 135: grammatically awkward. Once cannot isolate a sequence by PCR. One can amplify a nucleic acid by PCR or determine a sequence. I would suggest rephrasing

- Line 211: “ebolavirus” should be “Ebola virus”: hamster infections with any of the other four ebolaviruses have not been reported to my knowledge.

- Line 219, 220 etc.: should be “shared by members of the genera”.

- Line 226: should be “nucleotide sequence identity”

- Line 262-263: should be “reactivity of cuevavirions, ebolavirions, and marburgvirions”

- Line 279: which implications are that? Could the authors elaborate?

- Lines 281-289: I would suggest deletion as the paragraph just repeats previous statements.

6. References:
- Line 306, line 346: italicize “_Filoviridae”
- Line 328: should be “Ebola virus” (Ebola capitalized)
- Line 338: italicize “_Castor”

7. Figures 1-2
- At the very least, all species names should be italicized (and in Figure 1, the correct species name “_Marburg marburgvirus” should be used). However, this would still be incorrect, as sequences of individuals were used for comparison, which belong to species. Consequently, the trees should list viruses (and not virus species) and animals (and not animal species). Ideally would be both, as in “Lloviu virus (_Lloviu _cuevavirus)” and “North American deermouse (_Peromyscus _maniculatus _bairdii)”. The filovirus names are “Ravn virus”, “Marburg virus”, “Lloviu virus”, “Sudan virus”, “Reston virus”, “Ebola virus”, “Bundibugyo virus” and “Tai Forest virus”.

8. Figure 3 and elsewhere: see above. Should be “sequence in the meadow vole (_Microtus…) etc.

9. Figure 6: should be “of the shared insert of members of the other species”

10. Figure 7: should be “filovirid-like gene inserts”

11. Table 1, Legend: should be “where E stands for ebolavirus, L for Lloviu virus, C for cricetid filovirid-like, and M for marburgvirus sequences”

---

## Round 0.2 · accepted · Accept

· Academic Editor

Accept

Thanks for making all the stylistic changes suggested by the reviewers.